# Effect of COVID-19 Vaccination Campaign in Belgian Nursing Homes on COVID-19 Cases, Hospital Admissions, and Deaths among Residents

**DOI:** 10.3390/v14071359

**Published:** 2022-06-22

**Authors:** Sara Dequeker, Milena Callies, Catharina Vernemmen, Katrien Latour, Laura Int Panis, Romain Mahieu, Lennert Noppe, Muhammet Savsin, Els Duysburgh

**Affiliations:** 1Department of Epidemiology and Public Health, Sciensano, 1050 Brussels, Belgium; milena.callies@sciensano.be (M.C.); catharina.vernemmen@sciensano.be (C.V.); katrien.latour@sciensano.be (K.L.); laura.intpanis@sciensano.be (L.I.P.); elza.duysburgh@sciensano.be (E.D.); 2Department of Infectious Disease Prevention and Control, Common Community Commission, Brussels-Capital Region, 1040 Brussels, Belgium; rmahieu@ccc.brussels; 3Agency for Care and Health, Infection Prevention and Control, Government of Flanders, 1030 Brussels, Belgium; lennert.noppe@vlaanderen.be; 4Direction de la Recherche, de la Statistique et de la Veille des Politiques, AVIQ, 6061 Charleroi, Belgium; muhammet.savsin@aviq.be

**Keywords:** COVID-19, nursing homes, surveillance, epidemiology, Belgium

## Abstract

In view of the grave situation during the first two waves of SARS-CoV-2 virus (severe acute respiratory syndrome coronavirus-2), nursing homes (NHs) were prioritised for vaccination once vaccines became available in Belgium. The aim of this study was to assess the effect of the COVID-19 (Coronavirus Disease 2019) vaccination campaign on COVID-19 cases, hospital admissions, and deaths among residents living in Belgian NHs. All 1545 Belgian NHs were invited to participate in a COVID-19 surveillance program. In Belgium, before vaccination, COVID-19 morbidity and mortality rates were driven by the situation in the NHs. Shortly after the COVID-19 vaccination campaign, and later the booster campaign, the number of hospital admissions and deaths among NH residents dropped, while clear peaks could be observed among the general population. The impact of vaccination on virus circulation was less effective than expected. However, due to the high vaccination coverage, NH residents remain well protected against hospital admission and death due to COVID-19 more than one year after being vaccinated.

## 1. Introduction

The SARS-CoV-2 virus (severe acute respiratory syndrome coronavirus-2) has spread rapidly across the world, with almost 520 million cases and 6.3 million deaths by the second week of May 2022 [1]. Older age (65 years and older) has been associated with greater risk of developing more severe disease and death [2,3,4], leading COVID-19 (Coronavirus Disease 2019) to affect elderly in long-term care facilities considerably [4,5], due to the multiplicative effects of the residents’ age, health frailty, and infection risk [5]. Several studies have reported a decline in hospital admissions and deaths due to COVID-19 among elderly and nursing home (NH) residents after vaccination [6,7,8,9,10].

In Europe, Belgium has one of the highest numbers of NH beds per 1000 population and has the highest percentage of people 65 years and older and 80 years and older living in residential long-term care facilities [11]. In 2020, 19% of the Belgian population was older than 65 years of age, of which 5.3% lived in a NH [12,13].

In view of the grave situation during the first two COVID-19 waves, NHs were prioritised for vaccination once vaccines became available in Belgium. Between 28 December 2020 and 24 March 2021, 89.4% of NH residents received the primary course of the Pfizer-BioNTech vaccine [14]. Between 6 October and 31 December 2021, a booster with a mRNA COVID-19 vaccine (Pfizer-BioNTech and Moderna) was offered to all NH residents. On 18 January 2021, the vaccination campaign started among the general population targeting priority groups (e.g., health care workers, people > 65 years old, and individuals having at least one predefined comorbidity) before being extended to the whole population [15,16]. In November 2021, it was decided that the entire Belgian population aged 18 and above who had received a primary vaccination course would be invited to receive a booster dose [16]. On 24 March 2021, the end of the vaccination campaign among NH residents, 4.4% of the Belgian general population had received a primary vaccination course. This increased to 34.1% on 26 June 2021 (end of the third wave) and to 73.6% on 4 October 2021 (start of the fourth wave). On 1 May 2022, 79.3% of the Belgian general population had received a primary vaccination course and 61.9% had also received a first booster dose [17,18,19].

The aim of this study was to assess the effect of the COVID-19 vaccination campaign on COVID-19 cases, hospital admissions, and deaths among residents living in Belgian NHs.

## 2. Material and Methods

A detailed description of the study design and implementation was published in Eurosurveillance and Archives of Public Health [20,21]. As described in these papers, in March 2020, a comprehensive COVID-19 surveillance program in NHs was put in place to report COVID-19 confirmed and possible cases, hospital admissions, and deaths. A confirmed case was defined as a SARS-CoV-2 infection confirmed by a positive polymerase chain reaction or rapid antigen test conducted in a laboratory, irrespective of clinical signs and symptoms. A possible case was defined as a COVID-19 infection diagnosis based on clinical presentation with or without a positive compatible computed tomography thorax [20]. All 1545 Belgian NHs, representing 143,887 residents, were invited to report data at least once a week. Before mid-May 2020, around 60% of the NHs participated on a weekly basis in the surveillance program; from mid-May 2020 to July 2021, the number of NHs participating on a weekly basis was around 95%, and after July 2021, between 60% and 75%.

To demonstrate if differences in hospital admission and death due to COVID-19 among NH residents before and after vaccination were statistically significant, we compared the proportion of hospital admissions and deaths due to COVID-19 among NH residents with a confirmed COVID-19 infection during the first and second COVID waves (before vaccination campaign) with these proportions occurring during the three waves following the vaccination campaign. We used a chi square test to assess if differences between these proportions were statistically significant.

## 3. Results

In Belgium, during the first and second wave of the COVID-19 pandemic, morbidity and mortality rates were driven by the situation in the NHs (Figure 1). This period accounted for 87.6 COVID-19-related deaths among 1000 NH residents and 0.8 COVID-19-related deaths among 1000 members of the general population (excluding NH residents), thus indicating a 100-times higher mortality rate among NH residents compared to the general population. Although they represent only 1.3% of the Belgian population, 57.2% of the COVID-19-related deaths were NH residents during these first two waves.

Later in 2021 and 2022, three additional COVID-19 waves (third, fourth, and fifth waves) were observed. The third wave lasted approximately four months, with a peak in April 2021; the fourth wave lasted three months, with a peak in December 2021; and the fifth wave lasted less than two months, with a peak beginning in February 2022. Whereas during the third and fourth wave, clear peaks in cases, hospital admissions, and deaths were observed in the general population, no such peaks were observed among NH residents (Figure 1). In the meantime, the delta (B.1.617.2) variant of SARS-CoV-2 became dominant in June 2021, and concerns about vaccine effectiveness grew [22,23]. At the start of the fourth wave, which coincided with the start of the booster campaign in the NHs, the numbers of cases among residents increased but dropped quickly and never reached the level of cases reported in the general population. Hospital admissions and mortality among NH residents increased only slightly during the fourth wave, not comparable with the levels observed before the vaccination campaign and among the general population. At the end of December 2021, the omicron (B.1.529) variant became dominant, marking the start of the fifth wave, which resulted in an increase in the number of cases among NH residents and among the general population [23]. However, COVID-19-related hospital admissions and deaths among NH residents remained low, and were not comparable with the levels observed before the vaccination campaign.

Since the third wave until 20 March 2022, COVID-19-related deaths among NH residents and among the general population (excluding NH residents) dropped to 10.9/1000 NH residents (8 times less deaths than before the NH vaccination campaign) and 0.6/1000, respectively. After vaccination, 18.3% of all COVID-19-related deaths were NH residents compared to 57.2% before the start of the vaccination campaign. Before vaccination, 15.1% of the NH residents per COVID-19 confirmed infections were hospitalised and 32.5% died compared with respectively 4.7% and 4.9% after vaccination (*p* < 0.0001 for both comparisons).

During the first wave, due to limited availability of COVID-19 tests, 18% of the reported cases in the NH were confirmed cases. This changed when COVID-19 tests became widely available, and from mid-June 2020 until mid-March 2022, 61% of reported cases were confirmed cases.

It should be noted that in the NH surveillance program, all residents with a positive test who are hospitalised are considered as a confirmed COVID-19 hospital admission (including screening). For hospital admissions among the general population, only patients with COVID-19 as a main diagnosis are included as a confirmed COVID-19 hospital admission (excluding screening).

## 4. Discussion

In Belgium, we observed a decline in hospital admissions and deaths due to COVID-19 among NH residents after COVID-19 vaccination. Immediately after the vaccination and booster campaign, the number of hospital admissions and deaths among NH residents dropped, while peaks continued to be observed among the general population, who had a lower rate of COVID-19 vaccination and/or booster coverage than the NH residents at the same moment in time. This indicates an effect of COVID-19 vaccination campaigns in Belgian NHs on COVID-19 cases, hospital admissions, and deaths among NH residents. As far as the authors know, this is the first study to describe this in Belgium.

At the start of the COVID-19 crisis, in the Belgian NHs, different infection prevention and control (IPC) measures were implemented at regional and local levels, such as quarantine measures, ventilation, cohorting of cases, use of additional personal protective equipment, and preventive and rapid testing [20]. From the end of February 2021 onwards (end of vaccination campaign in NHs), these IPC measures were slowly relaxed. Among others, under certain circumstances, visitors were again allowed, and residents could again leave the NHs [24,25]. This relaxation of IPC measures did not result in an increase in deaths and hospital admissions among NH residents during the third Belgian COVID-19 wave, although we observed an increase in deaths and hospital admissions among the general population, who had not yet been vaccinated on a large scale. This observation stresses the effect of the COVID-19 vaccination campaign in Belgian NHs on hospital admissions and deaths.

After the vaccination campaign in NHs, different variants of SARS-CoV-2 occurred and raised concerns about vaccine effectiveness over time [22,26]. Due to its higher transmissibility [27], the omicron (B.1.529) variant resulted in a steep increase in the number of cases among NH residents and among the general population in Belgium and Europe [28]. However, after the occurrence of these variants, the number of hospital admissions and deaths among Belgian NH residents remained substantially lower compared to the levels observed during first two waves, before the start of the vaccination campaigns. Although it is known that different variants have different effects on hospitalization and death due to COVID-19, until recently, the additional effect of vaccination remains clear for these variants [29]. This is also the case for the omicron variant, where a steep increase in cases did not result in an increase in hospital admissions proportional to previous waves, and where the vaccination effect remained [30]. In the United Kingdom, they found that booster vaccines reduced the possibility of hospital admissions due to an infection with omicron by 68% compared to individuals who were not immunized [31].

The beneficial effect of vaccination has also been illustrated via observations from the Netherlands. There, booster vaccination among NH residents started at the end of November [32], almost two months later than in Belgium. This resulted in a clear increase in mortality among NH residents during the fourth COVID-19 wave [33], an increase that was not observed in Belgian NHs.

Despite the clear decrease in hospital admissions and deaths after vaccination, NH residents continue to be more severely affected than the general population, although the differences between the two populations are less pronounced than before the vaccination campaigns. This stresses the need to remain vigilant about the evolution of morbidity and mortality among NH residents and to continue monitoring through surveillance programs.

## 5. Conclusions

Although the impact of vaccination on virus circulation was less effective than hoped and/or expected, and despite the occurrence of new SARS-CoV-2 variants of concern, due to a successful vaccination campaign with high coverage, NH residents remain well protected against COVID-19-related hospital admission and death more than one year after being vaccinated.

## Figures and Tables

**Figure 1 viruses-14-01359-f001:**
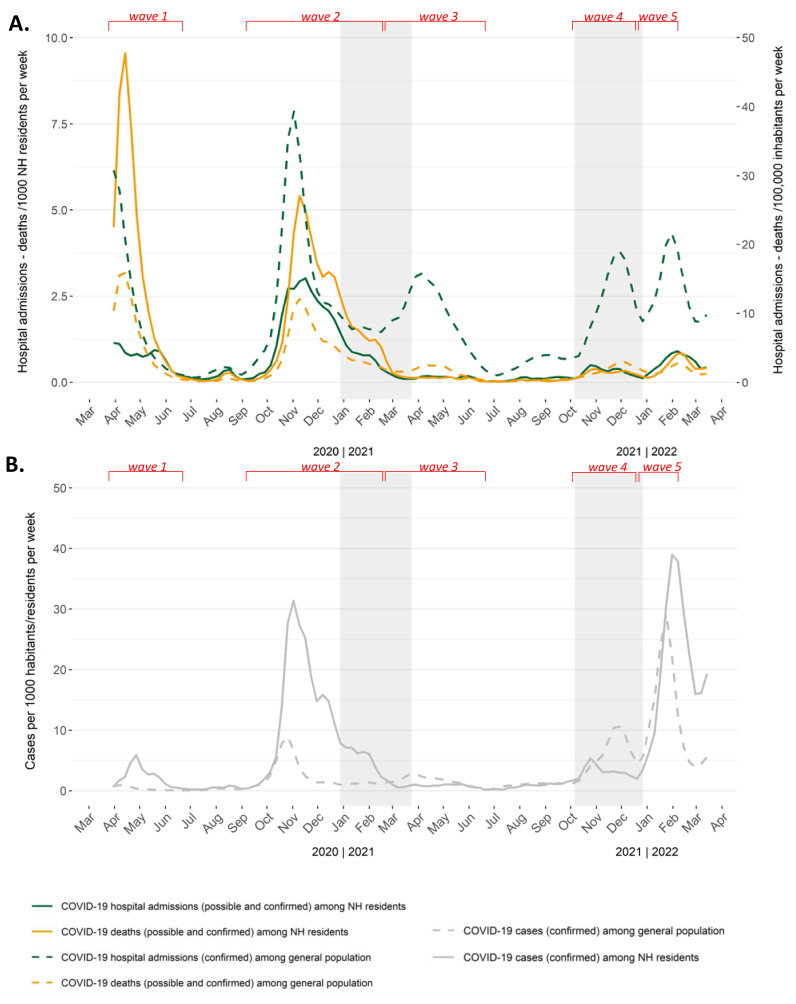
(**A**) Hospital admissions and deaths per 1000 nursing home (NH) residents (left *y* axis) and hospital admissions and deaths per 100,000 inhabitants (including NH residents) (right *y* axis), per week (two-week moving average), 6–20 March 2022. (**B**) Confirmed cases per 1000 nursing home (NH) residents and confirmed cases per 1000 inhabitants (including NH residents), per week (two-week moving average), 6–20 March 2022. Grey boxes indicate the COVID-19 (Coronavirus Disease 2019) vaccination campaign and the booster administration in NHs. Red brackets indicate the different waves.

## Data Availability

Data used to calculate cases and hospitalization among NH residents and to calculate cases and hospitalization among the general population are available at (https://epistat.wiv-isp.be/covid/, accessed on 21 June 2022). The data used to calculate the mortality are not available through open data source but are available on reasonable request. The statistical codes that support the findings of this study are available from the corresponding author on reasonable request.

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
