# Peer review of "Effect of COVID-19 Vaccination Campaign in Belgian Nursing Homes on COVID-19 Cases, Hospital Admissions, and Deaths among Residents"

_viruses, 2022, doi:10.3390/v14071359_

Round 1

Reviewer 1 Report

The manuscript of viruses-1680364 studied the effects of COVID-19 vaccination and the booster in Belgian nursing homes on COVID-19 cases, hospital admissions and deaths among residents living in Belgian NH. There are some suggestions here:
1. The authors concluded that "due to the high vaccination coverage, NH residents remain well protected against hospital admission and death due to COVID-19 more than one year after being vaccinated." About the numbers of hospital admission and death decreasing, besides vaccination, there are some other factors such as different SARS-CoV-2 variants and the controlling and protecting measures adopted to prevent the spread of COVID-19 for NH. The authors should consider and discuss the combined effects contributing to the decreased hospital admission.
2. In the introduction, there was little information about the related reports about situation of COVID in NH for other countries. And also the authors should explain the current status of the related studies and specify the significance of the study.
3. If available, comparing the impact in Belgium to that in other countries.
4. The authors compared the NH residents to general people as for the numbers of hospital admission and death decreasing due to the vaccination and gave the percentage of vaccination for NH residents but not the general people in Belgium. Without the information of the COVID-19 vaccination for the general people in Belgium, how did the authors compare the impact of the vaccination?

Author Response

Thank you for your comments. Please see the attachment (page 2 - 4) for our response. 

Reviewer 2 Report

In this study, the authors collected a series of SARS-CoV-2 data from nursing homes in Belgium, and compared these numbers, including confirmed case numbers, morbidity and mortality rates before and after the vaccination and booster campaigns, with the data from general populations. The authors concluded that COVID-19 vaccination efficiently protected the nursing home residents. Overall, the manuscript is clearly written and the data is very vivid that should be of interest to a broad audience. A few issues should be addressed prior to publication:

  1. Many comparisons have been made throughout this manuscript, however, without employing statistical tests. To make these data more objective and reliable, please employ statistical analysis and provide p values for these comparisons. To do so, I’d also suggest marking the 1st-5th waves separately (see point 5) and calculating the p values for each wave and for the period outside of these waves as a negative control.
  2. As the authors stated, “during the third and fourth wave clear peaks in cases, hospital admissions and deaths were observed in the general population, no such peaks were observed among NH residents”. The authors concluded that this is due to vaccination campaign. Is there any possibility that policies in nursing homes in Belgium got updated after vaccination campaign and also contribute to this decreased morbidity and mortality rates? For examples, maybe a more restrictive visiting policy made it much harder for SARS-CoV-2-infected visitors to enter nursing homes. Consistently, a low confirmed case number among NH residents during the third wave was observed. Thus, more should be discussed here.
  3. Since the confirmed cases among NH residents are low in third and fourth waves but high in fifth waves, more analysis and comparison should be focused on the fifth waves.
  4. This study mainly focused on Belgium, what about other countries? Is there any related/similar data available for other countries? If some comparison between countries could be made, that would be more interesting.
  5. I’d suggest combining Figure 1 and 2 together and putting them in parallel. This would facilitate readers understanding and make it easy for comparison.
  6. In Figure 1, please directly mark the 1st-5th waves separately, and also label the vaccination campaign and booster administration, so that the readers can more easily understand this figure.
  7. Introduction part could be improved for more information for nursing home residents and situations in Belgium.

Author Response

Thank you for your comments. Please see the attachment (page 4 - 6) for our response.

Reviewer 3 Report

It’s a quite simple study of COVID-19 vaccination campaign on COVID-19 cases in both nursing homes and general population.  The results shown vaccination well protected nursing homes nursing homes from hospital admission and death. The paper is well written, and results were clearly presented.

Author Response

Thank you for your comments. Please see the attachment for our response.

Round 2

Reviewer 1 Report

The authors made the responses to all the comments and suggestions.